# Hyperammonaemia in Dogs Presenting with Acute Epileptic Seizures—More than Portosystemic Shunts

**DOI:** 10.3390/ani15172558

**Published:** 2025-08-30

**Authors:** Sara M. Fors, Sarah Østergård Jensen

**Affiliations:** AniCura Referral Animal Hospital Bagarmossen, Ljusnevägen 17, Bagarmossen, S-128 48 Stockholm, Sweden; sarah.jensen@anicura.se

**Keywords:** epilepsy, hyperammonaemia, hepatic encephalopathy, neurology, seizures

## Abstract

This retrospective multicentre study investigated the frequency of hyperammonaemia and hepatic encephalopathy in dogs with acute seizures and if ammonia concentration can be temporarily elevated because of seizure activity. The medical records from ten veterinary hospitals were reviewed, and 58 dogs were found that had recent or ongoing seizures and had ammonia analysis performed within 24 h. Ten dogs of fifty-eight had hyperammonaemia and three of them concurrent documented liver disease, and in two cases had portosystemic shunts. In seven dogs, no definitive diagnosis was achieved. This study suggests that hyperammonaemia during or after a seizure can occur in dogs without obvious acute liver failure or portosystemic shunts. However, this study could not confirm that elevated ammonia concentration was a consequence of seizure activity.

## 1. Introduction

Epilepsy is a common neurological disorder in dogs characterised by recurrent seizures. The reported prevalence in dogs ranges between 0.5% and 5.7% [1,2]. As part of the diagnostic approach, the International Veterinary Epilepsy Task Force suggest the analysis of ammonia [3]. Hyperammonaemia has long been described as one of the primary causes of seizures, for example, in hepatic encephalopathy (HE) [4,5,6,7,8,9,10,11,12]. In addition to seizures, a range of neurological symptoms can result from HE and vary from mild behavioural abnormalities, cognitive impairment, tremor, and ataxia to coma [5,13,14]. Although seizures can occur with HE, these would be part of a collection of clinical symptoms rather than taking place in isolation [14].

In HE, levels of ammonia in the brain have been reported to increase due to abnormal ammonia metabolism in astrocytes, and glial–neuronal communication is affected [7,8,9,10,11,12]. Ammonia is lipophilic and passes freely over the blood brain barrier (BBB) even under normal circumstances [5,14], but in conditions with elevated blood ammonia concentrations, an increased amount can enter the brain and result in astrocyte dysfunction and swelling, which are considered to be central in the pathogenesis of ammonia neurotoxicity [14].

Glutamine is considered to play a main role in the pathogenesis [7,11,14,15]. Neurons take up glutamine released from the astrocyte and the enzyme glutaminase converts glutamine to the excitatory neurotransmitter glutamate. Glutamate is then released into the synaptic cleft where astrocytes rapidly reuptake it. In hyperammonaemia, glutamine increases intracellularly in astrocytes in the brain, leading to oxidative damage to the mitochondria and the inhibition of key enzymes of the TCA cycle (Krebs cycle). This disrupts cell metabolism, resulting in astrocyte swelling, as well as increased activation of NMDA receptors [7,14,15]. However, an experimental study by Thrane et al. [13] indicates failure of potassium buffering in astrocytes as a crucial mechanism in ammonia neurotoxicity, resulting in triggering cortical neuronal disinhibition and seizures.

Ammonia is formed in the body as a result of protein metabolism by the deamination of amino acids. The production of ammonia occurs predominantly from cellular metabolism of glutamine, a major circulating amino acid, in addition to urea metabolism. Ammonia production primarily involves the intestines, where enterocytes convert glutamine to glutamate, with minor contributions by urease-producing bacteria and bacterial protein degradation [14], and ammonia metabolism also includes the kidneys, muscle, brain, and liver [5,16,17,18,19]. The role of the kidneys in ammonia metabolism is complex: they contain both glutaminase and glutamine synthetase and are capable of both the metabolism and synthesis of glutamine [5,14]. Metabolic acidosis, as well as alkalosis and hypokalaemia, affect ammonia metabolism in the kidneys. Skeletal muscle myocytes can temporarily remove ammonia by conversion to glutamine and act as a buffer [14].

Detoxification of ammonia to urea occurs primarily in the liver through a series of biochemical reactions involving several enzymes, known as the urea cycle, and urea is excreted by the kidneys into the urine. Elevated blood ammonia concentrations result either from increased production of ammonia or decreased elimination [17,20,21,22]. In liver insufficiency, astrocytes are forced to detoxify a higher amount of ammonia [7]. However, the liver has a large reserve capacity, and even in the face of severe hepatic insufficiency, ammonia detoxification is often maintained. In portosystemic shunting, high-ammonia-concentration blood in the portal vasculature can directly reach systemic circulation [14].

Hyperammonaemia in dogs is mainly associated with hepatic encephalopathy caused by portosystemic shunting, either due to congenital portosystemic shunts or acquired portosystemic collateral vessels due to portal hypertension [5,14]. Lidbury [5] reported that acute liver failure (ALF) without portosystemic shunting can lead to severe changes in ammonia metabolism but concluded that this appears to be a less common cause of hyperammonaemia in dogs. In a retrospective study including 49 dogs [23] with acute liver failure, ammonia was analysed in 11/49 dogs and was increased in 7. No clear diagnostic criteria of ALF exist in the veterinary literature [6,23], and ALF was defined in a study by Lester et al., 2016 [23], as the acute onset of clinical signs with concurrent identification of hyperbilirubinaemia and coagulopathy.

Other reported causes of hyperammonaemia in humans and dogs include urea cycle enzyme deficiencies, organic acidurias, azotaemia, gastrointestinal haemorrhage, urinary retention, urinary infections with urease-producing bacteria, acute myeloblastic leukaemia, chronic myelocytic leukaemia, chemotherapy, metabolic causes like hyperinsulinaemic hypoglycaemia, distal renal tubular acidosis, primary carnitine deficiency, and fatty acid oxidation defects [5,17,24,25,26,27,28,29,30,31,32,33,34].

Acidosis is reported as a cause of hyperammonaemia in humans [35], and both Sato et al. [36] and Nakamura et al. [35] demonstrated a strong correlation between higher ammonia values and acidosis in humans with idiopathic epilepsy, in addition to extensive muscle contractions. In cardiopulmonary arrest or haemorrhagic shock without muscle contractions, red blood cells can produce ammonia through acidosis, leading to hyperammonaemia [37].

As part of the diagnostic approach to determining the aetiology of seizures, the International Veterinary Epilepsy Task Force proposes to include fasting and post-prandial bile acids (BA) and/or ammonia (NH3) analysis in addition to minimum database blood (MDB) tests and urinalysis [3]. The main purpose of analysing ammonia is to diagnose hepatic encephalopathy, as hyperammonaemia is associated with hepatic disease due to the central role of the liver in ammonia metabolism. Suggested MDB blood tests include complete blood cell count (CBC) and a serum biochemistry profile including sodium, potassium, chloride, calcium, phosphate, alanine aminotransferase (ALT), alkaline phosphatase (ALP), total bilirubin, urea, creatinine, total protein, albumin, glucose, cholesterol, and triglycerides.

Transient hyperammonaemia following seizures has been described in humans [35,36,38,39,40,41,42,43] and recently also in cats [44]. This temporary increase in blood ammonia concentration, without concurrent signs of liver failure, is known as post-ictal hyperammonaemia and it is considered to result from intense muscle contractions during seizures [35,36,38,39,40,41,43,44]. Aetiologies of seizures in humans with reported transient hyperammonaemia included intracranial lesions, endocrine disease, and idiopathic epilepsy [36,43]. The association between seizures and transient post-ictal hyperammonaemia is not fully understood, but several mechanisms are suggested in the literature.

Normal muscle activity produces ammonia, particularly during intense exercise or muscle contractions [16,17,45,46,47]. During motor seizures, especially generalised tonic-clonic seizures, the intense and prolonged muscle contractions can lead to increased ammonia production that may overwhelm the liver’s capacity to metabolise ammonia, resulting in hyperammonaemia. Extensive contractions of muscles during a generalised motor seizure produce ammonia through deamination of adenosine monophosphate in the purine nucleotide cycle and branch-chained amino acid (BCAA) deaminase reactions [18,36]. Another possible explanation could be reduced blood flow during seizures that further impair the liver’s ability to metabolise ammonia, exacerbating hyperammonaemia [18,45].

In humans, patients with generalised tonic-clonic seizures exhibited significantly higher ammonia levels compared to those with focal motor seizures. Patients in status epilepticus had even higher ammonia levels [39]. This likely reflects the total amount of recruited muscle activity. Nakamura et al. [35] stated that a transient elevation of ammonia levels can occur as a result of seizures caused by idiopathic epilepsy alone. Results by Tomita et al. [42] also demonstrated hyperammonaemia during generalised seizures, further supporting this.

Hung et al. [39] reported an incidence of transient post-ictal hyperammonaemia of 67.77% in their study population of humans admitted for seizure activity, with a reduction in ammonia levels within 7.78 h. In a human case series, ammonia levels had normalised in three out of six patients within 2–3 h [38]. In a study by Sato et al. [36], the incidence of transient post-ictal hyperammonaemia in humans was 48.3%. In contrast, Nilsson et al. [44] reported that 22% of the cats included in their study had post-ictal hyperammonaemia.

Transient hyperammonaemia in a dog with seizures has been described in one case report [48]. A 3-year-old German Shepherd presented with severe generalised seizures and a basal ammonia concentration within the reference range. An ammonia stimulation test revealed severe hyperammonaemia. Hepatic disease, portocaval shunting, urea cycle enzyme deficiencies, drug therapy, and urinary tract obstruction were ruled out. The dog was diagnosed as protein-intolerant and had hyperammonaemia after stimulation that persisted for at least six weeks and then spontaneously resolved. The findings in the case report do not support the theory of seizures causing hyperammonaemia in the dog, and to the authors’ knowledge, there are no published studies with confirmed hyperammonaemia as a result of seizures in dogs.

The distinction between hepatic encephalopathy/elevated ammonia concentrations due to hepatic dysfunction and post-ictal hyperammonaemia is important, as in the latter case the phenomenon is self-limiting and does not require any treatment or extensive investigations. In addition, post-ictal hyperammonaemia is not necessarily related to adverse outcomes, which makes it an important differential diagnosis, especially in comparison to hepatic dysfunction and HE [39].

The aim of this study was to retrospectively investigate the prevalence of hyperammonaemia and hepatic encephalopathy in dogs presenting with acute epileptic seizures. Secondarily, this study sought to evaluate whether transient post-ictal hyperammonaemia due to seizure activity occurs in dogs, as reported in humans and cats.

## 2. Materials and Methods

The case records of dogs presented between the 1 March 2014 and 15 December 2024 to ten AniCura Veterinary Hospitals in Sweden (Regiondjursjukhuset Bagarmossen, Kalmarsund, Läckeby, Jönköping, Västra, Hässleholm, Falu, Kumla, Jägarvallen, and Norsholm) were reviewed. Data was collected from the patient management system by searching for ammonia analysis, dog, and the following diagnose codes: “seizure(s)”, “epileptic seizure(s)”, and “epilepsy”.

Inclusion criteria were complete information regarding signalment (age, breed, weight, and sex), description and characterisation of seizures (status epilepticus, cluster seizures, and single seizure at presentation or within hours of admission), MDB blood samples consisting of haematology (haematocrit, erythrocytes, total count white blood cells, and platelets) and biochemistry profile (total protein and/or albumin, urea and/or creatinine, glucose, alanine aminotransferase, and alkaline phosphatase), and analysis of ammonia performed within 24 h after the last reported seizure. Twenty-four hours was chosen as the maximum time limit, as the purpose was to evaluate ammonia concentrations in dogs with recent and/or ongoing seizures. In dogs with an ammonia concentration exceeding the upper reference limit, only those with concurrent bile acid analysis, bile acid stimulation test, or analysis of bilirubin reported in the record were included.

According to the human literature, transient post-ictal hyperammonaemia persists for a limited period of time [39,43]. Yanagawa et al. [43] reported that ammonia was normalised the following day in 47% of their study population, and Hung et al. [39] concluded that median plasma ammonia concentrations had decreased within 3–8 h, with an average interval of 466.79 min. The timing of ammonia sampling in the dogs was therefore further divided into two groups: one where the sampling and analysis of ammonia was performed within 8 h and one after 8 h but within 24 h.

If available, information regarding diagnostic imaging, ultrasound of the liver, or computed tomography (CT) with hepatic angiography to evaluate portosystemic shunting (PSS) was also reviewed, and, when available, additional results of other modalities like magnetic resonance imaging (MRI) of the brain were examined. Additionally, bicarbonate and/or lactate concentrations were also noted if analysed. Information from follow-up visits was also included when available.

In accordance with previous studies investigating hyperammonaemia in association with seizures in the absence of hepatic disease in humans and cats, dogs where MDB analyses and bile acids and/or the bile acid stimulation test or bilirubin analysis were within the reference interval were considered unlikely to have hepatic failure [6,23,35,36,38,39,40,41,42,43,44]. In addition, adult dogs without ongoing treatment with liver-enzyme-inducing medication were required to have normal ALP values [49] (pp. 655–658). If the bile acid stimulation test or post-prandial bile acids were within the normal range, PSS was considered unlikely [38,41,43,44].

Seizures were classified as focal or generalised according to the 2017 International League Against Epilepsy paper on the classification and terminology of seizure types [50] based on the description of the seizure activity in the record. Cluster seizures are characterised as two or more seizures within 24 h with normalised mentation between seizures and status epilepticus as ongoing seizure activity lasting five minutes or longer or two or more seizures within 24 h without normalised mentation between seizures [51].

For the analysis of ammonia in dogs admitted to AniCura Regiondjursjukhuset Bagarmossen, clearly defined standard operating procedures (SOPs) for both pre-analytical and analytical procedures were applied to minimise errors. Since haemolysis can artefactually increase ammonia concentration, haemolytic samples were not analysed [52,53,54].

Venous blood was collected in EDTA tubes and immediately chilled, separated in a cooled centrifuge at 5 °C at 1500 G, and EDTA-plasma analysed within 30 min on a Cobas^®^ C311 (Roche Diagnostics Limited, Rotkreuz, Switzerland). A two-point determination method was used, utilising an enzymatic reaction including glutamate dehydrogenase (GLDH) where the end result is L-glutamate + NADP^+^ + H_2_O. The concentration of NADP^+^ produced is directly proportional to the ammonia concentration, which is determined by measuring the decrease in absorbance [53,55]. The measurement range for ammonia with this method is 10–1000 µmol/L and the reference interval (RI) is 14–54 µmol/L [56]. The RI was determined by in-house studies conducted prior to implementation of the instrument at the laboratory.

In the dogs from the other nine AniCura Veterinary Hospitals, blood samples were immediately centrifuged following collection, and heparinised plasma was analysed within 1 h on a Catalyst Dx (IDEXX Nordics, Solna, Sweden) with the use of dry-slide technology. This technology involves a chemical reaction between ammonia and bromophenol blue, where the resulting colour change is read optically [57]. With this method, the measurement range for ammonia is 0–950 μmol/L and the reference interval (RI) for ammonia is 0–99 μmol/L for puppies and 0–98 μmol/L for adults, according to the manufacturer [58].

## 3. Results

The medical records of 267 dogs that had a seizure diagnosis and blood ammonia analysis reported were examined. After reviewing the records, 209 dogs were excluded. Reasons for exclusion were that either the ammonia sample was not analysed in connection to a recent seizure, it was impossible to determine when the last seizure activity occurred in relation to ammonia sampling, or seizure activity was either excluded or not possible to classify according to the taskforce criteria after reviewing the description in the record.

Fifty-eight dogs fulfilled the inclusion criteria (Figure 1). Thirty-two different breeds were represented: fourteen mixed breed, five Chihuahuas, four French Bulldogs, two Miniature Pinschers, two Yorkshire Terriers, two Miniature Poodles, two Border Collies, two Maltese, two Pugs, and one of each of the breeds Alpine Dachsbracke, American Toy Fox Terrier, Australian Shepherd, Cavalier King Charles Spaniel, Chinese Crested Dog, Coton de Tuléar, Danish-Swedish Farm Dog, Doberman, English Springer Spaniel, Golden Retriever, Groenendael, Kleiner Münsterländer, Nova Scotia Duck Tolling Retriever, Papillon, Pomeranian, Rottweiler, Russkiy Toy, Shetland Sheepdog, Schiller Hound, Småland Hound, Staffordshire Bull Terrier, Wachtelhund, and Whippet. Twenty-six entire males, twenty-four entire females, five male neutered, and three female neutered dogs were included. Their age ranged from 2 to 164 months; the mean was 42 months and median 19.5 months. The weight ranged from 0.96 to 50.5 kg, the mean was 7.1 kg, and the median 12.11 kg.

Hyperammonaemia was demonstrated in 10/58 (17%) dogs. Three (5.2%) of those, two female Miniature Pinschers, 4 and 164 months old, respectively, and one female Border Collie, 28 months old, had confirmed hepatopathy based on laboratory testing and/or diagnostic imaging. Ammonia concentrations were 106, 142, and 325 µmol/L (reference interval 14–54 µmol/L, Cobas^®^ C311). Two (3.4%) of these dogs had portosystemic shunts that were subsequently surgically corrected, and both presented in ongoing status epilepticus.

A summary of the results is presented in Figure 1.

The other seven (12%) dogs with non-hepatic-related hyperammonaemia were two Chihuahuas, one Rottweiler, one Groenendael, one Papillon, one Yorkshire Terrier, and one mixed-breed dog. Age varied from 2 to 66 months. Three were males, one a neutered male and three females. Ammonia concentrations in these dogs ranged between 58 and 240 µmol/L (58, 73, 75, 84, 101, 112, and 240 µmol/L), with a reference interval 14–54 µmol/L, Cobas^®^ C311.

One of these dogs had minimally elevated ALT, and one had mildly elevated fasting bile acids. Four dogs, three growing puppies and one adult dog previously treated with corticosteroids, had mildly elevated ALP values.

None of the seven dogs with hyperammonaemia without confirmed hepatopathy had documented seizures or other neurological symptoms prior to presentation. Consciousness was impaired on arrival in 6/7 dogs with non-hepatic hyperammonaemia and in in 2/3 dogs with hyperammonaemia of hepatic origin.

One of the three dogs with confirmed hepatopathy, nr. 9, had inter-ictal intermittent impaired mentation and bizarre behaviour before diagnosis and had surgical correction of an intra-hepatic PSS.

Ammonia concentration was analysed within eight hours in 25 dogs and after eight hours but within 24 h in 33 dogs. In the ten dogs where hyperammonaemia was demonstrated, six had ammonia analysed within 8 h and four after 8 h. Four of the seven dogs with hyperammonaemia without confirmed hepatopathy had ammonia concentrations analysed within 8 h and three after 8 h but within 24 h. Detailed information is available in Table 1 and Appendix A.

A detailed description and diagnostic values for each of the 10 dogs with hyperammonaemia where ammonia was analysed <24 h after seizures are shown in Appendix A.

## 4. Discussion

The current study aimed to establish clinically relevant information on the prevalence of hyperammonaemia and hepatic encephalopathy and the possibility of transient post-ictal hyperammonaemia in dogs presenting with acute epileptic seizures, which could be used to aid the clinical reasoning approach in such cases.

In this study, 58 dogs with epileptic seizures had ammonia analysed within 24 h of seizure activity. Ten of these dogs (17%) had hyperammonaemia, where three dogs (5.2%) had documented hepatic encephalopathy and two of those (3.4%) suffered from PSS. For the remaining seven dogs (12%), no clear cause for the hyperammonaemia could be established, although PSS was ruled out in three dogs, and no other indications for ALF could be found on either clinical investigations or blood samples in the remaining four.

This finding is in contrast to the current veterinary literature reporting that hyperammonaemia mainly is associated with HE and most often occurs as a result of shunting of portal blood in portosystemic shunts [5,14,17,25] but can also, although less frequently, result from acute liver failure (ALF) [5,6,23].

The results from this study indicate that post-ictal hyperammonaemia can occur in the absence of acute hepatic failure or PSS in dogs. Although some findings indicated that the hyperammonaemia possibly could be caused by seizure activity, the results could not confirm that elevated ammonia levels were indeed caused by seizure activity alone or transient in the seven dogs where hepatic dysfunction was ruled out.

Gow [14] reported that seizures rarely occur in isolation in dogs with HE but in conjunction with other neurological symptoms. This correlates well with the results of this study. Consciousness was impaired on arrival in 6/7 dogs with non-hepatic hyperammonaemia and in in 2/3 dogs with hyperammonaemia of hepatic origin. As all dogs were presented in close proximity after or during ongoing seizure activity, it is difficult to distinguish between impaired consciousness due to hyperammonaemia or a post-ictal state.

However, of the ten dogs with elevated ammonia in this study, only one dog (nr. 9) had other neurological inter-ictal symptoms, specifically intermittent bizarre behaviour and impaired mentation, consistent with HE. This dog was diagnosed with an extra-hepatic portosystemic shunt, which was later surgically corrected. None of the other nine dogs had a history of or displayed other neurological symptoms. However, as some symptoms like cognitive impairment can be subtle, we cannot exclude that they may have gone unnoticed by the owner. High ammonia concentrations in a dog with recent seizure activity without a history of other inter-ictal neurological symptoms should raise the suspicion of another possible aetiology than HE, and the clinician is advised to repeat ammonia analysis and/or perform bile acid stimulation testing before pursuing extensive investigations for PSS.

Elevated ALP without concurrent ALT elevation and simultaneously normal bile acids/bilirubin concentration is not a typical presentation of ALF in dogs, and the mildly increased ALP values noted in the three youngest dogs were interpreted as within the normal range for a growing puppy [49] (pp. 655–658). A previous report [23] defined ALF as the acute onset of clinical signs with concurrent identification of hyperbilirubinaemia and coagulopathy. Numerous definitions of ALF are described in humans, and most include acute onset of icterus [6]. None of the seven dogs with suspected non-hepatic-related hyperammonaemia had clinical signs of coagulopathy or icterus, bilirubin was normal in the three dogs where the analysis was performed, and bile acids was normal in the remaining four, making ALF less likely.

One dog (nr. 3) had mildly elevated pre-prandial bile acids but normal CT-angiogram. The final diagnosis in this dog was MUE (Meningoencephalitis Of Unknown Etiology). Follow-up ammonia and pre-prandial bile acid analysis was performed after six weeks, and ammonia concentration had decreased from 240 to 64 µmol/L (14–54) and bile acids from 40 to 39 µmol/L (<20–25). One possible explanation for this persistent slight elevation of ammonia could be occult gastrointestinal haemorrhage [17,26], as the dog had a history of gastrointestinal disease, but hepatic dysfunction could not be completely excluded due to the mildly abnormal bile acid result. Another dog (nr. 7) had minimally elevated ALT but normal ALP and bilirubin within the reference interval, making ALF less likely as the cause of hyperammonaemia. The dog died during admission without a bile acid test being performed.

None of the dogs had abnormalities in their white blood cell count indicating leukaemia and none were treated with chemotherapy; therefore, those aetiologies were excluded. Complete blood cell count should always be included in investigations in cases of non-hepatic hyperammonaemia, as well a thorough history regarding ongoing medication.

Due to the retrospective nature of the study, it was not possible to investigate the potential impact of metabolic defects as alternative contributors to hyperammonaemia. Urea cycle enzyme deficiencies, organic acidurias, primary carnitine deficiency, fatty acid oxidation defects, and distal renal tubular acidosis could not be excluded either, as specific diagnostic testing for these abnormalities was not performed in the included dogs. When hyperammonaemia of non-hepatic origin is encountered, especially in young dogs without convincing evidence of PSS or microvascular dysplasia, inherited metabolic disorders as well as distal renal tubular acidosis should be investigated.

Hyperammonaemia can also occur due to urinary retention or obstructive uropathy alone [28] or in combination with urinary tract infections with urease-producing bacteria [29,30,31,32,33,34]. None of the included dogs had clinical signs of urinary retention or obstruction. One dog (nr. 1) had concurrent *E. coli* cystitis and was treated with amoxicillin according to resistance testing. Ammonia was re-evaluated after 3 weeks and was within reference interval. Although infrequent, less than 1% of *E. coli* is capable of producing urease, according to Konieczena et al. [59]. This dog had no evidence of PSS (normal BA stimulation and CT-angiogram) and no clinicopathological findings supporting ALF. Therefore, urinary tract infection could be a potential cause of the hyperammonaemia in this dog. Urinary analysis, including bacterial culture, should be considered in dogs presenting with elevated ammonia concentrations, in particular when PSS or ALF is ruled out. The possibility of obstructive uropathy must also be taken into consideration in cases with hyperammonaemia of non-hepatic origin.

Azotaemia has also been described as a cause of hyperammonaemia in both humans and cats [17,60], but all seven dogs with hyperammonaemia without hepatic insufficiency had creatinine and urea within the reference interval, which makes azotaemia an unlikely cause of hyperammonaemia. It is important to evaluate dogs with hyperammonaemia regarding renal function and possible azotaemia, as decreased kidney function can increase blood ammonia concentration [5,14].

Acidosis is reported as a cause of hyperammonaemia in humans [35]. Sato et al. [36] and Nakamura et al. [35] demonstrated a strong correlation between increased ammonia concentration and acidosis in humans with idiopathic epilepsy, in addition to extensive muscle contractions. In cardiopulmonary arrest or haemorrhagic shock, red blood cells can produce ammonia through acidosis, leading to hyperammonaemia [37]. Bicarbonate was only analysed in 2/7 dogs and was within normal limits in both, thus not interfering with ammonia concentrations. For the remaining five dogs, it cannot be excluded that acidosis affected ammonia concentration. As acidosis and elevated lactate can affect ammonia concentrations, it is important to take bicarbonate and/or lactate values in consideration when interpreting hyperammonaemia and repeat ammonia measurement when acidos/lactic acidosis is corrected. Thus, the clinician could consider adding blood gas analysis whenever a dog with hyperammonaemia in the absence of ALF and HE is presented.

Disorders of the gastrointestinal tract, like haemorrhage, when excess nitrogen can override the liver’s excretory capacity [26], can affect the ammonia metabolism and thus blood ammonia concentrations. We cannot exclude gastrointestinal haemorrhage contributing to hyperammonaemia in the dogs in this study from the available information in the records. It is important to include gastrointestinal bleeding, also in occult form, as a potential differential diagnosis in cases with hyperammonaemia. Hyperammonaemia is also described in cats with nutritional disorders like cobalamin and arginine deficiency [60]. Whether this is true also for dogs is currently unknown. In future prospective studies, the analysis of cobalamin could be included to investigate cobalamin deficiency as a possible cause for hyperammonaemia in seizures in dogs.

In a study by Sato et al. [36], the incidence of transient post-ictal hyperammonaemia in humans was 48.3%. Hung et al. [39] reported an incidence of transient post-ictal hyperammonaemia of 67.77% in their human study population, with a reduction in ammonia concentration within 7.78 h. In a human case series, ammonia had normalised in three out of six patients within 2–3 h [38]. Nilsson et al. [44] reported that 22% of the cats included in their study had post-ictal hyperammonaemia. If seizure-associated post-ictal hyperammonaemia occurs in dogs and ammonia concentration decreases within 2–8 h, as reported in most humans [38,39], transient post-ictal hyperammonaemia could have gone undetected in 33/58 (57%) of dogs in this study due to delayed ammonia analysis. Another limitation was the lack of follow-up analysis of ammonia in close proximity, making it impossible to evaluate whether hyperammonaemia was transient or persistent. We suggest that elevated ammonia concentrations, especially within 8 h after seizures in dogs, particularly in cases without other neurological symptoms or clinical signs of ALF, should be repeated to confirm or exclude transient post-ictal hyperammonaemia before further diagnostic testing for PSS is pursued.

Possible pre-analytical errors causing false high ammonia values include haemolysis in the sample, inadequate chilling of the sample, exposure to smoke from surroundings and staff handling the sample, and prolonged time from sampling to analysis (more than 30 min) [52,53,54]. However, pre-analytical error is considered unlikely in this study since analyses were performed by trained biomedical scientists and standard operating procedures were followed, including criteria for rejection of the sample.

As this is a retrospective study, it has several inherent limitations. Some of the main limitations include a small study group, selection bias regarding case groups, and information bias, as data quality is dependent on the accuracy of historical medical records, which may be incomplete or inconsistently documented. Information is limited to the variables available in medical records, and it is difficult to control for all relevant variables retrospectively. This can make it difficult to prove causality, but associations can still be made.

There was a selection bias with a high proportion of young dogs, often of small breeds, in cases where portosystemic shunt was suspected or in dogs where previous liver disease was suspected based on history. Of the seven dogs with hyperammonaemia after seizures without definitive evidence of acute liver pathology or PSS, five were younger than 8 months, one was 13 months, and one 66 months. It is therefore possible that the results are not applicable to the general population of dogs.

Other limitations of the study include the lack of standardised work-up to exclude primary liver pathology; the timing of ammonia sampling, diagnostic imaging, and minimum database blood tests; a predetermined sampling time from the last seizure to analysis of ammonia, and a lack of follow-up ammonia analysis for the evaluation of possible transient elevation. In several cases, information from the case records did not clearly state whether bile acids were pre- or post-prandial. The same was true for ammonia. As post-prandial values usually are higher than fasting ones [61], it cannot be excluded that, at least to some extent, ammonia values were influenced by post-prandial effects. Future studies should include information about whether ammonia concentrations are fasting values or post-prandial to facilitate the interpretation of ammonia levels.

Another important limiting factor is that it was not possible to reach a definitive diagnosis in the majority of dogs due to the limited data available, as the cases are historical. However, it was demonstrated that hyperammonemia can be seen in absence of HE and ALF which, considering the clinical implications of a HE or PSS diagnosis, is relevant for the planning of further diagnostic investigations and the prognosis for the patient. Thus, the result of this study warrants future prospective studies to further investigate and characterise a possible transient post-ictal hyperammonaemia.

Future prospective studies with a more varied cohort are needed, along with standardised testing protocols to further investigate the association between hyperammonaemia and seizure activity in dogs as well as other underlying aetiologies. In addition to MDB blood tests, ammonia, bile acid stimulation and blood gas analysis should be included, as well as cobalamin/B12. Follow-up sampling of ammonia should also be conducted in a standardised manner, especially with regard to the timing of blood sampling, to investigate whether transient post-ictal hyperammonaemia occur in dogs. To further evaluate hepatic pathology, histopathology of the liver could be included in future studies. The investigation of urea cycle enzyme deficiencies, organic acidurias, distal renal tubular acidosis, primary carnitine deficiency, and fatty acid oxidation defects could also be considered.

## 5. Conclusions

According to the results of this multicentre retrospective study, 10/58 (17%) dogs presenting with recent or ongoing seizures where ammonia was analysed within 24 h had hyperammonaemia. Only three of these dogs had documented hepatic pathology, two (3.4%) of which had surgically confirmed and corrected PSS, and seven dogs had post-ictal hyperammonaemia without convincing clinicopathological evidence of acute hepatic dysfunction or portosystemic shunting. The findings indicate a possibility that non-hepatic hyperammonaemia seen with acute seizures in dogs is more prevalent than previously described in the veterinary literature.

Hyperammonaemia during or after epileptic seizures, especially when there are no other clinicopathological signs of decreased liver function or other neurological symptoms, should alert the clinician to consider other potential pathologies than portosystemic shunts or acute hepatic failure, and follow-up ammonia analysis should be performed before routinely starting an extensive investigation of possible PSS. Another important consideration is that since elevated ammonia concentrations in dogs with seizures can be seen with other aetiologies than liver insufficiency and PSS, it is not automatically associated with grave illness, and the prognosis can be good. The results of this study also indicate that standardised, prospective studies are warranted to investigate and characterise potential transient post-ictal hyperammonaemia in dogs and to further guide clinical decision making for patients presenting with seizures in clinical practice.

## Figures and Tables

**Figure 1 animals-15-02558-f001:**
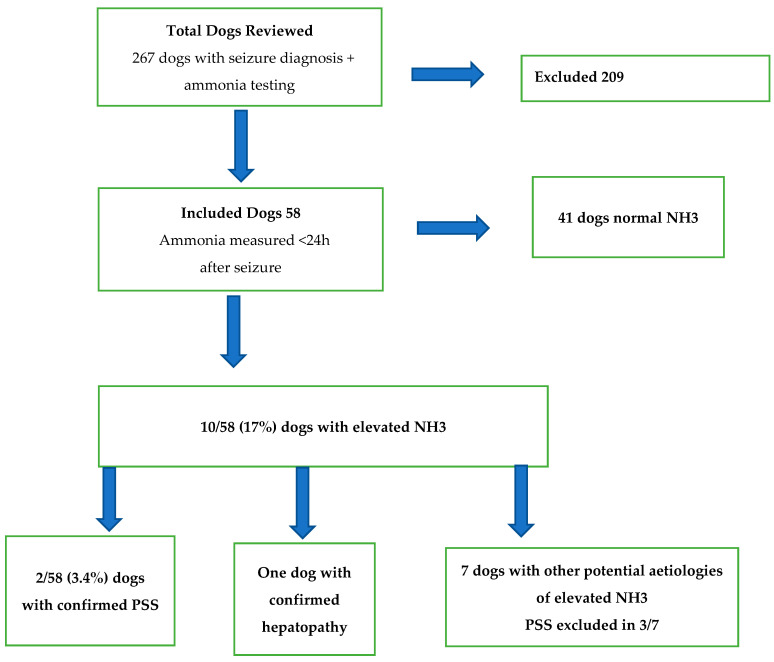
Flow chart with results of 58 dogs with seizures and ammonia concentration analysis. NH3 = ammonia. PSS = portosystemic shunt.

**Table 1 animals-15-02558-t001:** Time of analysis of ammonia of 10 dogs with seizures and elevated ammonia concentration. PSS = portosystemic shunt, ALF = acute liver failure.

	Breed	Ammonia Value at Presentation (RI 14–54 µmol/L)	Ammonia Analysed <8 h After Seizure	Ammonia Analysed >8 h <24 h After Seizure	Cause of Hyperammonaemia
**1**	Groenendael	112	Within 6 h	X	Possibly urease-producing *E. coli* No evidence of PSS or ALF
**2**	Rottweiler	73	Within 6 h	X	No definitive cause identified No evidence of PSS or ALF
**3**	Chihuahua Longhair	240	X	Within 12 h	No definitive cause identified No evidence of PSS, unlikely ALF
**4**	Yorkshire Terrier	75	Within 8 h	X	No definitive cause identified No evidence of ALF
**5**	Papillon	101	X	Within 12 h	No definitive cause identified No evidence of ALF
**6**	Mixed breed	58	X	22 h	No definitive cause identified No evidence of ALF
**7**	Chihuahua Longhair	84	Within 6 h	X	No definitive cause identified No evidence of ALF
**8**	Border Collie	106	X	>12 h <24 h	Suspected re-canalisation or persistent shunting after PSS surgery causing elevated BA and ammonia, alone or in combination with motor activity during seizure and elevated lactate
**9**	Miniature Pinscher	325	3 h	X	PSS
**10**	Miniature Pinscher	142	2.5 h	X	Hepatopathy

## Data Availability

The complete datasets used and/or analysed during the current study are available from the corresponding author on reasonable request.

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
