# Peer review of "Hyperammonaemia in Dogs Presenting with Acute Epileptic Seizures—More than Portosystemic Shunts"

_animals, 2025, doi:10.3390/ani15172558_

Round 1
Reviewer 1 Report
Comments and Suggestions for Authors The aim of this study was "to retrospectively investigate whether hyperammonaemia occur in dogs presenting with acute epileptic seizures, and which aetiologies that are asociated with elevated ammonia concentration in dogs with seizures. " However, since hepatic encephalopathy is among differential diagnoses of seizures, there is no need to investigate whether hyperammonaemia occur in dogs presenting with acute epileptic seizures, because the answer would be "yes".Further more, as fully described in discussion of this article. The study is very limited in investigating aetiology due to the nature of retrospective study, group size included, data limitations, of past cases. This limitation was obviously showed when authors did find hyperammonaemia cases without evidence regarding liver insufficiency but cannot perform further investigation. In addition, it's hard to distinguish post-ictal hyperammonaemia from concurrent hyperammonaemia by only testing ammonia level after seizure.
Therefore it is difficult to identify significant value of this article in terms of fulfilling its purpose, other than knowing there is possibility of hyperammonaemia due to unknow aetiology. I recommend improve research design or adding further investigation before publish a paper with strong argumentation and clear conclusion.
Author Response
Please see PDF-attachment

Reviewer 2 Report
Comments and Suggestions for Authors
The study titled “ Hyperammonaemia in dogs presenting with acute epileptic seizures-more than portosystemic shunts” set out to examine the association between seizures and elevated ammonia levels in dogs, with a special focus on non-hepatic causes of hyperammonaemia.
The introduction section asks to detail the link between epilepsy and hyperammonaemia; in fact, it is necessary to understand what the mechanisms are. It is also asked to better explain what the clinical effects related to this condition may be. In this regard, given the structuring of the manuscript, it is recommended to first explain ammonia metabolism and causes, and then discuss diagnostic tests. Also explain why it is necessary to distinguish between hepatic and post-ictal hyperammoniemia in epileptic dogs. The materials and methods section is well structured, the methods on ammonia analysis including instruments and timing were well detailed. What stands out is that it is a single text, it could be improved with a division into paragraphs, for example one part on biochemical analysis, one on statistics. The authors feel free with respect to the various paragraphs to be created but a breakdown is still requested. Authors report with respect to sample collection that 24 hours have passed, but time and/or temporal variability is not explained. The discussions discuss several potential causes of hyperammonaemia, but there is a lack of insight into the implications of these causes in dogs, and the various clinical implications need to be added. Could you be more explicit about the limitations? A final summary of your findings should also emerge in the discussions.
Reviewer 3 Report
Comments and Suggestions for Authors
The topic is very relevant for the field of epileptology. The authors have studied the frequency of hyperammonaemia in dogs with acute seizures, potential causes, and if ammonia concentration can be temporarily elevated because of seizure activity.
The methodology is very modern and complex, authors using modern methods of blood biochemistry, ultrasound of the liver or computed tomography (CT) with hepatic angiography to evaluate portosystemic shunt (PSS), and, when available, additional results of magnetic resonance imaging (MRI) of the brain.
The results have revealed that post-ictal hyperammonaemia can occur in the absence of acute hepatic failure or PSS in dogs. Although some findings indicated that the hyperammonaemia possibly could be caused by seizure activity, results could not confirm that elevated ammonia levels were caused by seizure activity alone or transient in dogs where hepatic dysfunction was ruled out. These results could be transferable to human health.
The conclusions are consistent with the evidence and arguments presented.
The references are very relevant, including also some relevant author’s previous experience in the field.
I suggest some small corrections. For Introduction and Discussions authors may see also:
1.Ștefănescu, R.A.; Boghian, V.; Solcan, G.; Codreanu, M.D.; Musteata, M. Electroencephalographic Features of Presumed Hepatic Encephalopathy in a Pediatric Dog with a Portosystemic Shunt—A Case Report. Life 2025, 15, 107, DOI, 10.3390/life15010107
2.M. ArmaÅŸu, R.M.A. Packer, S. Cook, G. Solcan, H.A. Volk, An exploratory study using a statistical approach as a platform for clinical reasoning in canine epilepsy, The Veterinary journal, 2014, 202, 292-296, DOI: 10.1016/j.tvjl.2014.08.008
